# OpenReview forum: "Improving Context-Aware Preference Modeling for Language Models"
_NeurIPS.cc/2024/Conference — NeurIPS 2024 poster_

### Official Review · Reviewer_rZtt · 2024-07-09

**Soundness:** 3
**Presentation:** 4
**Contribution:** 4
**Rating:** 6
**Confidence:** 4

**Summary:**

The paper focuses on fine-tuning LLMs to improve their ability to handle context-aware preference modeling. The authors address the challenge of the underspecified and ambiguous nature of natural language preferences by introducing a two-step modeling process. This includes selecting a context and evaluating preferences within that context. The approach is backed by the introduction of new datasets, named RPR, which are designed to test the effectiveness of context-specific preference modeling. The study provides extensive experimental evidence showing how context-aware models can surpass traditional methods in handling ambiguous and context-specific scenarios.

**Strengths:**

1. The paper tackles a crucial issue in the realm of language modeling by enhancing the LLM's ability to understand and process user preferences in context-dependent scenarios. This is particularly important as LLMs are increasingly used in diverse real-world applications.
2. The introduction of the RPR datasets is a notable contribution, as these datasets specifically aim to disentangle context-specific preferences from general preferences, offering a valuable resource for further research.
3. The paper provides thorough experimental results that not only demonstrate the effectiveness of the proposed method but also explore various aspects of context-aware preference modeling, showing improvements over existing models like GPT-4 and Llama 3.

**Weaknesses:**

1. The paper could benefit from testing the proposed method across a broader range of general benchmarks, such as MMLU and AGI-Eval, to assess how fine-tuning for context-aware preferences might affect the LLM's original capabilities or general applicability.
2. It remains unclear how applicable the proposed method is to other LMs, especially those of different sizes or from different series. Addressing this would help validate the robustness and versatility of the method.

**Questions:**

Please see my concerns in weaknesses.

---

> ### Author Rebuttal · Authors · 2024-08-06
>
> Thank you for your detailed review and feedback. Please find our responses below.
>
> &nbsp;
>
>
> > The paper could benefit from testing the proposed method across a broader range of general benchmarks, such as MMLU and AGI-Eval, to assess how fine-tuning for context-aware preferences might affect the LLM's original capabilities or general applicability.
>
> We agree that training generative models with context-aware rewards is a great direction, and part of our broader agenda. In particular, we believe this could improve sensitivity to the system prompts and diverse users, and to prompts with multiple instructions or constraints, which is something we have observed state-of-the-art models struggle with. The design decisions here are not obvious though (where does context for MMLU come from, for example, especially given its more objective nature) and add another layer of complexity that we believe would be better suited for future work.
>
> &nbsp;
>
> > It remains unclear how applicable the proposed method is to other LMs, especially those of different sizes or from different series.
>
> This is a good point. We took the opportunity to use RPR to finetune one of the stronger 2B parameter reward models according to the Reward Bench leaderboard, which is based on Gemma (hf:Ray2333/Gemma-2B-rewardmodel-baseline). We used the same hyperparameters as used for the 7B Mistral RM in the paper. We obtained the following context-conditioned results, which we will include the paper (Mistral results included for reference, best Gemma model bolded):
>
>
> | || Gemma-2B-NC |  Gemma-2B-CTX  | **Gemma-CARM** | \| |Mistral-7B-CTX | Mistral-CARM |
> | --------|--------|:-: | :-: | :-: | :-: | :-:| :-: |
> | RPR Criteria || 0.511 | 0.761 |  **0.968** | \| |0.867 | 0.985 |
> | RPR Scenarios || 0.511 | 0.655 |  **0.909** | \| |0.749 | 0.962 |
> | Multifaceted Bench ||  0.508 | 0.597 |  **0.681** | \| |0.679 | 0.787 |
> | Preference Bench || 0.852 |  **0.861** |  0.849 | \| |0.915 | 0.919 |
> | HHH (CTX) || 0.751 |  0.751 |  **0.760** | \| |0.905 | 0.919 |
> | Rewardbench (CTX)| | 0.718 | 0.735 |  **0.786** | \| |0.833 | 0.871 |
> | Chatbot Arena (CTX*) || 0.745 |  0.806 |  **0.873** | \| |0.859 | 0.909 |
>
> Note: Multifaceted Bench is a new context-conditioned preference dataset — see general response.
>
> We observe that finetuning the Gemma RM shows similar patterns to finetuning the Mistral RM, and that the finetuned model is in many cases competitive with the larger Mistral base model.

---

### Official Review · Reviewer_t1sE · 2024-07-11

**Soundness:** 2
**Presentation:** 3
**Contribution:** 3
**Rating:** 5
**Confidence:** 4

**Summary:**

This paper divides preference modeling into two steps: first estimating the user's intent then evaluate the generated text within the context of this intent. The paper makes the Reasonable Preference Reversal Datasets, which encompass criteria and scenarios for preference data. Experiments find that models can achieve higher performance when providing intent contexts during evaluation.

**Strengths:**

- The approach of first estimating user intent before evaluation may be promising.
- This work constructs the first open-source context-conditioned preference datasets, which may be useful for future work.

**Weaknesses:**

1. Lack of citations and comparisons to related work. For example, Li et al. [1] generate the criteria for evaluation and then get the final answer according to the criteria, which is similar to this work.

2. In Table 2, the author aims to demonstrate that fine-tuning a context-aware reward model significantly enhances context-specific performance. However, the observed improvement in Table 2 could be attributed to the model being trained on data from the same distribution as the test set. Even for new datasets that are not context-specific, training the model on such datasets will likely improve its performance on its test sets.

3. The lack of experiments illustrating that two-step preference modeling achieves better performance compared to traditional reward modeling on general preference datasets is a concern. In Table 4, the model is prompted by context generated by GPT-4, leading to unfair comparisons with similar models. To demonstrate that two-step preference modeling is superior, it is essential to show its advantages for models of equal ability.

[1] Li J, Sun S, Yuan W, et al. Generative judge for evaluating alignment[J]. arXiv preprint arXiv:2310.05470, 2023.

**Questions:**

Why is two-step preference modeling necessary? The user's intent is inherently contained in the query, and traditional preference modeling implicitly evaluates whether the response aligns with the user's intent. Therefore, why should the estimated intent be treated independently? Additionally, ambiguity exists in estimating intent, as different individuals may have inconsistent interpretations of intent for the same query.

**Limitations:**

Please see Weakness part.

---

> ### Author Rebuttal · Authors · 2024-08-06
>
> Thank you for your detailed review and feedback. Please find our responses below.
>
> &nbsp;
>
>
> > citations /  Li et al. [1]
>
> Thank you for bringing Li et al. to our attention. We agree it is closely related and will add it to the related work, and we will revisit our literature search for other recent work we may have missed.
>
> As part of our rebuttal, we have run their Auto-J model, but found that both Ultra RM and the base Mistral RM outperform it on every dataset/benchmark we use, both unconditioned and conditioned. We note that Auto-J, including the datasets used in the Auto-J paper, seeks to directly address the unconditioned preference modeling problem, even if Auto-J generates criteria as part of its judging process.
>
> &nbsp;
>
>
> >Table 2 could be attributed to the model being trained on data from the same distribution as the test set.
>
> We agree this is true for RPR and will make this more clear in our paper (see, e.g., the stars in the new data ablation table of the PDF attachment). However, we note that our finetuned CARM also improves performance on all other context-specific datasets, including the newly added Multifaceted Bench (see General Response), as is now more clearly shown in Figure 1 of the PDF attachment.
>
> &nbsp;
>
>
> > In Table 4, the model is prompted by context generated by GPT-4, leading to unfair comparisons with similar models
>
> We disagree that these comparisons are unfair. In particular, *all* models—not just our CARM—have access to the *same* additional context, hence their improved performance relative to the “NC” column.
>
> &nbsp;
>
>
> > The lack of experiments illustrating that two-step preference modeling achieves better performance compared to traditional reward modeling on general preference datasets is a concern.
>
> We agree it would be ideal to show that two-step preference modeling can achieve better performance on general preference datasets. However, this requires two parts: (1) better context-specific modeling, and (2) strong context inference or specification. While we show in this work how we can improve the context-specific modeling problem, the context inference problem remains difficult. We have tried to generate context with respect to unconditioned queries as part of our work as well, and found that getting current models to generate useful context is quite challenging (for evidence of this, see the Auto-J results noted above), which is why we believe one of the next steps for future work is to investigate how to best obtain strong context *supervision* (as we allude to at L186-187). We think supervision of some kind is critical in order to go beyond the “user's intent is inherently contained in the query” as you wrote.
>
> Furthermore, we believe two-step preference modeling to be useful even if it could not improve on current unconditioned preference datasets such as Reward Bench. See next response.
>
> &nbsp;
>
> > Why is two-step preference modeling necessary? … why should the estimated intent be treated independently?
>
> We think your last sentence here captures our motivation: “ambiguity exists in estimating intent, as different individuals may have inconsistent interpretations of intent for the same query.” Part of our argument is that intent need not be estimated solely from the prompt—indeed, it can be:
>
> - specified for annotators (see L84-91), or via default rules similar to Constitutional AI, or via the system prompt
> - inferred from past interactions in cases of persistent context (e.g. user profiles, L287; see revised Table in the attached pdf), or
> - learned from data via some context supervision system.
>
> To the extent that there are inconsistent interpretations of the same query, any such context will usefully disambiguate the query, which provides several advantages:
>
> - **(a)** Given context, we reduce reliance on unstated, implicit assumptions made by annotators, thereby increasing overall agreement (see Ziegler at al., quoted at L88).
> - **(b)** Using context we can improve steerability and pluralistic alignment (see Sorensen et al.).
> - **(c)** Making the context explicit may also be useful to diagnose errors made by models.
> - **(d)** Notably, making context explicit allows us to change the aggregation rule from a Borda count (see Siththaranjan et al.) to a more flexible Social Welfare Aggregation (as in Bakker et al., who find this to be effective as a consensus mechanism).
>
> In the last case, (d), agreement with respect to unconditional preferences gathered from a group (and thus, performance on general preference datasets) seizes to become a good measure, as Equation (2) [L158-159] (or modifications of Equation 2 for non-EU SWFs) will not hold, so that explicit contextualized aggregation (of human preferences) becomes a better alignment target than overall preference (see Siththaranjan et al. for a concrete example of this).

---

> ### Comment · Reviewer_t1sE · 2024-08-11
> **Response to Authors**
>
> Thank you for the authors' detailed response that solves some of my questions through additional experiments and explanations.
>
> However, my core question still exists, which is ``whether two-stage modeling is better than traditional reward modeling''. I understand that using experiments to illustrate this can be difficult, but it should be an important part.
>
> In real conversation and preference annotation scenarios, the intent of a query is ambiguous and has multiple possibilities. The two-stage approach may lead to better modeling results through decomposition, but it may also result in error accumulation.
>
> Moreover, as Weakness1 pointed out, previous work has already explored developing evaluation criteria before conducting evaluations. This results in a limited contribution of this paper.
>
> I think the author's research direction is promising, but there are still many points that can be improved. So, I keep my rating.

---

### Official Review · Reviewer_mLs1 · 2024-07-14

**Soundness:** 2
**Presentation:** 2
**Contribution:** 2
**Rating:** 4
**Confidence:** 4

**Summary:**

This paper points out that the preference label can be reversed by inserting additional context into the prompt. Based on this observation, the authors build a paired preference dataset. The authors also try to provide some theoretical analysis.

**Strengths:**

* This paper studies a specific and interesting problem in preference data.
* The proposed data augmentation method may benefit preference optimization of LLMs and inspire further research.

**Weaknesses:**

My major concern lies in the experiments. The current experiments do not demonstrate the general advantage of using a paired dataset with reversed preference labels. On widely used preference datasets, the model trained on the proposed dataset does not outperform the baselines. It only performs better on the test set, which is built in the same way as the training set. Note that the preference labels on this test set are not verified by humans and could be unreliable.

**Questions:**

See weaknesses.

**Limitations:**

This paper does not have a limitation section.

---

> ### Author Rebuttal · Authors · 2024-08-06
>
> Thank you for your detailed review and feedback. Please find our responses below.
>
> &nbsp;
>
>
> > The current experiments do not demonstrate the general advantage of using a paired dataset with reversed preference labels.
>
> Thank you. This is a good criticism. We realized this post submission and have prepared a data ablation that finetunes the base Mistral reward model on context-conditioned data from other distributions, including:
>
> - Preference Collection (PC) (hf:prometheus-eval/Preference-Collection), in which the context is highly correlated with unconditioned preference, and
> - a “one-sided” version of our RPR datasets, where kept only one of the two sides by having GPT-4 pick one of the two criteria or scenarios that it thought was more suitable for the given prompt.
>
> The results are shown in Table 1 in the PDF attachment, and demonstrate the advantage provided by our RPR dataset. We will add this to our paper. Notably, the Multifaceted Bench (MF), released after the NeurIPS submission deadline, introduced an additional context-conditioned preference benchmark that is outside of our training distribution, where you can see that training on RPR strongly improves context-conditioned performance over the base model (and the PC model).
>
> &nbsp;
>
>
> > On widely used preference datasets, the model trained on the proposed dataset does not outperform the baselines.
>
> If you are referring to the no context versions of HHH, Reward Bench, and Chatbot Arena, these are unconditional preference benchmarks, where it is not expected that our model, trained for context-conditioned preference queries, would do better than the base model. We report results here mainly to provide a reference figure, and also to show that fine tuning on RPR does not hurt the unconditioned preference prediction of the base model.
>
> &nbsp;
>
>
> > It only performs better on the test set, which is built in the same way as the training set.
>
> This is not true. Our model not only performs better on the RPR test sets, but also: Multifaceted Bench (MF, see general response), HHH with context, Reward Bench with context, and Chatbot Arena with Context*. This is more clearly shown in Figure 1 in the PDF attachment, which we will add to our paper as Figure 2. Please note the errata for GPT-4 results in the general response.
>
> &nbsp;
>
>
> > Note that the preference labels on this test set are not verified by humans and could be unreliable.
>
> Human validation of the dataset labels was done by the authors (blind / response orders and criteria/scenarios randomized). As the authors, we are the closest to the data, and are able to provide high quality labels. In our original submission we reported 100 total labels (50 for RPR Criteria and 50 for RPR Scenarios; different prompts for each) in Table 2 (pdf page 7). Since submission, we have added an additional 100 labels (50 for each Criteria and Scenarios), achieving a total agreement of 97% for RPR Criteria and 95% for RPR Scenarios (100 labels each). This gives 95% confidence intervals for human-RPR agreement as follows:
>
>
> | RPR Criteria    | RPR Scenarios |
> | :--------: | :-------: |
> | $$(0.937, 1)$$  | $$(0.907, 0.993)$$ |
>
> Some noise in human agreement is unavoidable. As noted in our paper (L73) there is significant inter-human disagreement on unconditioned queries, e.g., humans agree with each other only 65.7% of the time on AlpacaEval. Even the carefully curated MT-bench shows has agreement of only 81-82% on strict preference queries (Zheng et al., Table 5). Thus, our 90+% agreement between the human authors and the synthesized labels is quite high, likely as a result of the added context and multiple levels of filtering.

---

### Official Review · Reviewer_j5YU · 2024-07-14

**Soundness:** 2
**Presentation:** 3
**Contribution:** 2
**Rating:** 4
**Confidence:** 3

**Summary:**

The motivation behind this paper is to address the critical challenges of finetuning language models (LMs) from pairwise preferences due to the underspecified nature of natural language. Direct preference feedback is often uninterpretable, inconsistent, and difficult to provide, especially when multidimensional criteria are involved. These issues arise from incomplete instructions or the diverse backgrounds of the individuals providing the feedback. To tackle these challenges, the authors propose a two-step preference modeling approach: first, selecting a context to resolve under-specification, and second, evaluating preference with respect to the chosen context.

There are mainly 4 contributions the authors have made:

**Decomposition of Reward Modeling Error**: The paper introduces a method to decompose reward modeling error into two components: context inference error and context-specific reward modeling error. This decomposition supports the idea that supervising both context and context-specific preference could align models more effectively with diverse human preferences.

**Context-Conditioned Preference Datasets**: The authors contribute several novel datasets designed to investigate the ability of LMs to evaluate context-specific preferences. These datasets, termed "preference reversal" datasets, isolate context-specific capabilities by disentangling them from general preferences.

**Context-Aware Reward Model**: The paper demonstrates the development and finetuning of a context-aware reward model, showing that this model achieves performance comparable to or exceeding that of state-of-the-art models like GPT-4 and Llama 3 70B. The context-aware model also shows improved performance in context-specific tasks.

**Experiments and Benchmarking**: The authors conduct experiments to benchmark the context-specific performance of various models, highlighting that current models benefit from additional context but often fail to fully utilize it. Finetuning with the preference reversal datasets significantly enhances the models' context-specific performance.

**Strengths:**

This paper presents a novel approach to preference modeling in LMs by integrating context-specific evaluation. The key contributions include the introduction of context-conditioned preference datasets, the decomposition of reward modeling error, the development of a context-aware reward model, and comprehensive experiments to validate the approach.

**Weaknesses:**

## Under-supported claims

In section 3.3, the author included various claims that I believe to be interesting but only so when they are better-substantiated.

**Assumption about Cardinality** The hypothesis that the cardinality of the space of contexts is smaller than that of the space of possible completions given a prompt is stated without empirical or theoretical justification. This assumption is critical to the argument about data efficiency in context annotation versus preference annotation. Without evidence or a rationale to support this assumption, the argument remains speculative.

## Details of the dataset curation
The reviewer appreciate the author's efforts in curating the dataset.
Section 4 briefly mentions that the dataset was generated using GPT-4 Turbo with a series of prompts designed to maximize validity. However, it lacks specific details about the prompts, the criteria for selection, and the methodology for ensuring the validity of the samples. This ambiguity makes it difficult to assess the soundness of the curation process.

## Clarity
While the paper presents interesting ideas, there are some areas in writing and presentation that could benefit from improvement:
- The introduction and related work sections are detailed, but they could be more concise to better highlight the key contributions and their significance.
- The figures and tables, although informative, could be optimized to more effectively convey the main points. Some elements might appear redundant or unclear.
- Enhancing the narrative flow would help in making the logical progression of the arguments clearer and more accessible to the reader.

**Questions:**

## Annotation and intent inference

I particularly do not understand the assumption related to Bernoulli distribution:

Section 3.1 states that an annotator implicitly infers intent i from prompt x and samples a preference from a Bernoulli distribution. This process of intent inference by annotators is critical yet highly under-explained. There is no empirical evidence or studies cited that demonstrate annotators' ability to accurately infer intents from prompts. Additionally, the variability in annotators' interpretations and the potential biases they introduce are not addressed.

## Intent distribution

The paper asserts that both users and annotators may possess or infer a distribution of intents and that annotation for most preference queries involves a distribution rather than a specific intent. This is a substantial claim that needs empirical support.

## Data curation process
I would appreciate it if the authors can provide further details on data curation.  In particular, how the quality of the synthetic data is maintained.

## Human validation
What is the scale of this validation, and how are the human validators selected? Is the sample size sufficient to generalize the results to the entire dataset?

**Limitations:**

Yes.

---

> ### Author Rebuttal · Authors · 2024-08-06
>
> Thank you for your detailed review and feedback. Please find our responses below.
>
> &nbsp;
>
>
> > Under-supported claims / Assumption about Cardinality
>
> The framing of this in the discussion is not as “claims” but rather a “conjecture” (L182) and a “hypothesis” (L188,L193). As noted in the text, we do find “some” (L181,L193) empirical support for each, in Tables 2 and 5. We leave a more detailed exploration of this to future work, as the focus of this work is on improving context-specific preference modeling.
>
> &nbsp;
>
>
> > Data curation process / “lacks specific details about the prompts, the criteria for selection, and the methodology for ensuring the validity of the samples”
>
> As noted in the main text at L211 (right after introducing the datasets), the full details of the dataset curation process, including prompts, criteria for selection, and methodology for ensuring validity (e.g. L794), are detailed in Appendix B (L766-L815). If there is some shortcoming in the presentation there, we would appreciate a more specific critique.
>
> &nbsp;
>
>
> > Clarity / Enhancing the narrative flow
>
> We have added a Figure 2 and revised Figure 1 (see pdf attachment) to clarify the framework and main results. We believe the narrative flow is strong, but would appreciate and happily consider any more specific critiques the reviewer has on this point.
>
> &nbsp;
>
>
> > Assumption related to Bernoulli distribution / Intent distribution
>
> Modeling paired preference as sampling from Bernoulli distribution is standard; e.g., in the widely used Bradley-Terry model. Allowing for an intent distribution (as opposed to a single intent) seems like a natural modeling choice that broadens the scope of the model (it gets us Equation (3) in addition to the single intent Equation (4)). We do not believe these modeling choices require empirical justification, as we are not claiming in Sections 3.1-3.2 that real human annotators explicitly infer an intent or distribution of intents, or that real human annotators explicitly model their label as a Bernoulli distribution. Indeed, at L148-151 we adopt the (commonly used) Expected Utility model, which is used to arrive at Equations (3) and (4); but it is well known that humans systematically deviate from expected utility.
>
> Our formalism is Sections 3.1-3.2 is best understood as any other idealized model: it offers analytical insights. In our case, our model is used to arrive at the bounds in Equation (3) and (4), which expands on and strengthens the motivation in Section 2, and suggests the discussion in Section 3.3. As with any model, it is up to the reader to determine whether the model in question is a reasonable, and useful enough, approximation to reality.
>
> &nbsp;
>
>
> > There is no empirical evidence or studies cited that demonstrate annotators' ability to accurately infer intents from prompts. Additionally, the variability in annotators' interpretations and the potential biases they introduce are not addressed.
>
> This is a core motivation for our work, as set out at L82-85: “If we train models using non-basic preference annotations, the contextual biases and assumptions underlying those judgments may be implicitly embedded into the model [35, 48] … Rather than rely on annotators to integrate the correct distribution of contextual assumptions … ”, wherein it is suggested that we might not want to rely on annotators to “accurately infer intents” [i.e., contexts], as this is subject to “the variability in annotators' interpretations and the potential biases”. A descriptive study of the biases and assumptions made by real human annotators is beyond the scope of the present work.
>
> &nbsp;
>
>
> > Human validation
>
> The human validation of the dataset labels was done by the authors (blind / response orders and criteria/scenarios randomized – we will add this detail to the paper). As the authors, we are the closest to the data, and are able to provide high quality labels. In our paper we reported 100 total labels (50 for RPR Criteria and 50 for RPR Scenarios; different prompts for each). Since submission, we have added an additional 100 labels (50 for each Criteria and Scenarios), achieving a total agreement of 97% for RPR Criteria and 95% for RPR Scenarios (100 labels each). This gives 95% confidence intervals as follows:
>
> | RPR Criteria    | RPR Scenarios |
> | -------- | ------- |
> | $$(0.937, 1)$$  | $$(0.907, 0.993)$$  |

---

### Author Rebuttal · Authors · 2024-08-06

We thank the reviewers for their time, consideration, and numerous comments that will help us improve the manuscript. We have responded to each reviewer individually. If you find our rebuttal to be responsive to your concerns, we kindly ask you to consider recommending “accept” --- we believe context-specific modeling and pluralistic alignment is an important direction that has received recent interest from a number of different research groups, and that the present submission is complete and provides interesting results and contributions in this area.

We have the following general comments and revisions to note:

&nbsp;


**Added new Figure, and updated Table 6 to better present the results**

See PDF attachment (Figure 1 and Table 2). Figure 1 of the PDF attachment will appear as Figure 2 in our revised manuscript.

&nbsp;


**Updated to include results on Multifaceted Bench and discuss the concurrent work by Lee et al. [1]**

Post submission, the work [1] by Lee et al. was released, which includes a synthesized dataset of diverse system prompts for finetuning generative models. Although formed for a different purpose, their dataset includes multiple system prompts for the same user prompt, which allows it to be used in a similar fashion as our RPR datasets. We run context-specific evaluation on their dataset. The main results are now shown in the PDF attachment (Figure 1), and we will include a discussion of their work in the Related Works section.

[1] Lee, Seongyun, et al. "Aligning to thousands of preferences via system message generalization." arXiv preprint arXiv:2405.17977 (2024).

&nbsp;

**Updated to include a Data Ablation**

Reviewer mLs1 made the excellent point that “The current experiments do not demonstrate the general advantage of using a paired dataset with reversed preference labels.”

In response, we have prepared a data ablation that finetunes the base Mistral reward model on context-conditioned data from other distributions, including:

- Preference Collection (PC), in which the context is highly correlated with unconditioned preference by Kim et al. (2024) (Prometheus 2), and
- a “one-sided” version of our RPR datasets, where we keep only one of the two sides by having GPT-4 pick the criteria that it thinks is best suited for the given prompt.

The results are shown in Table 1 in the PDF attachment, and demonstrate the advantage provided by our RPR dataset. We will add this to our paper. Notably, on Multifaceted Bench (MF), which is outside of our training distribution, training on RPR strongly improves context-conditioned performance over the base model (and the PC model).

&nbsp;

**Errata re: GPT-4 results**

After submission, we noticed that the GPT-4 results appeared high (e.g. our submission reported a score of 93.5% on Reward Bench, which is far higher than what it obtains on the Reward Bench leaderboard). This was due to a bug in our code that was counting ties issued by GPT-4 as being correct.  **Only the results reported for GPT-4 Turbo were affected—all other originally reported results are correct.** Here are the corrected numbers for GPT-4, which are also reflected in the attached PDF Figure 1.


|          | RPR-C (CTX)   | RPR-S (CTX) | PB (CTX) | MF (CTX) | HHH (NC) | HHH (CTX) | RB (NC) | RB (CTX) | CBA (NC) | CBA (CTX) | CBA (CTX*) |
| --------| :--------: | :-------: | :--------: | :-------: |  :--------: | :-------: |  :--------: | :-------: |  :--------: | :-------: | :-------: |
| Original GPT-4 | 0.899 | 0.725 | 0.868 | - | 0.950 | 0.964 | 0.935 | 0.904 | 0.825 | 0.833  | 0.930|
| Corrected GPT-4 | 0.901 | 0.748 | 0.860 | 0.640 |  0.871 | 0.873 | 0.824 | 0.821 | 0.720 | 0.771 | 0.858 |

After this correction, our finetuned Mistral CARM is the best performing model not only on RPR (for which it is in distribution), but also: Multifaceted Bench, HHH (CTX), Reward Bench (CTX), and Chatbot Arena (CTX*). Note that GPT-4 does worse on average than Llama3-70B not because it is a worse at context-specific prediction necessarily, but because we do not have access to its logits so that its scores are restricted to an integer scale and often result in ties.

&nbsp;

**New context-aware finetune**

Reviewer rZtt made a good point that "it remains unclear how applicable the proposed method is to other LMs". As part of our rebuttal we finetuned a reward model based on Gemma 2B that performed relatively well for its size on Reward Bench. This shows that the benefit of training on RPR is not restricted to the Mistral 7B model we used in our submission. We report the results in our response to Reviewer rZtt below, which we will add to the paper.

&nbsp;

**Added sample from RPR**

We realized after submission that our manuscript did not include any samples from the dataset itself. We have since added a sample from RPR to the main text to help with the exposition. We are happy to share it during the extended discussion period if the reviewers would find it helpful.

---

### Comment · Ethics_Reviewer_CfH7 · 2024-08-10
**No ethical concerns**

This paper raises no ethical concerns

---

### Decision · Program_Chairs · 2024-09-25

**Decision:**

Accept (poster)

**Comment:**

The paper addresses the challenges of fine-tuning language models using pairwise preferences due to the underspecified nature of natural language. It proposes a two-step preference modelling approach, introducing context-conditioned preference datasets and developing a context-aware reward model.

The reviews are borderline and the authors suspect that one of the negative reviews contains artefacts indicating that it could have been generated by ChatGPT.

The paper introduces a two-step preference modelling approach that could potentially enhance the alignment of models with diverse human preferences (e.g. different domains for text generation have different rrequirements). The introduction of context-conditioning is a valuable contributioin and may be of benefit for downstream applications in model personalisation. The context-aware reward model demonstrates improved performance on context-specific tasks, with comprehensive experimental results supporting this claim.

In the rebuttal, the authors have provided additional data that counters the negative aspects identified by the reviewers. The rebuttal introduces additional experiments demonstrating the model’s advantages on newly introduced benchmarks, which somewhat mitigates the concerns raised in the first round of reviewing.

There is a significant discrepancy between reviewers mLs1 and j5YU , who lean towards rejection due to concerns about under-supported claims and experimental limitations, and reviewers rztt and t1sE, who see potential in the novel approach and datasets introduced.

The paper has the potential to influence the language model personalisation and preference modeling, particularly in how context is integrated into preference modelling. The introduction of context-conditioned datasets could serve as a valuable resource for future research.